# Three New Dihydrophenanthrene Derivatives from *Cymbidium ensifolium* and Their Cytotoxicity against Cancer Cells

**DOI:** 10.3390/molecules27072222

**Published:** 2022-03-29

**Authors:** Tajudeen O. Jimoh, Bruno Cesar Costa, Chaisak Chansriniyom, Chatchai Chaotham, Pithi Chanvorachote, Pornchai Rojsitthisak, Kittisak Likhitwitayawuid, Boonchoo Sritularak

**Affiliations:** 1Pharmaceutical Sciences and Technology Program, Faculty of Pharmaceutical Sciences, Chulalongkorn University, Bangkok 10330, Thailand; jimmmypeace@gmail.com (T.O.J.); brunocesarsoares@outlook.com.br (B.C.C.); 2Department of Biochemistry, Faculty of Health Sciences, Islamic University in Uganda, Kampala P.O. Box 7689, Uganda; 3Department of Biochemistry and Microbiology, Faculty of Pharmaceutical Sciences, Chulalongkorn University, Bangkok 10330, Thailand; cchoatham@gmail.com; 4Department of Pharmacognosy and Pharmaceutical Botany, Faculty of Pharmaceutical Sciences, Chulalongkorn University, Bangkok 10330, Thailand; chaisak.ch@chula.ac.th (C.C.); kittisak.l@chula.ac.th (K.L.); 5Natural Products and Nanoparticles Research Unit, Chulalongkorn University, Bangkok 10330, Thailand; 6Center of Excellence in Cancer Cell and Molecular Biology, Faculty of Pharmaceutical Sciences, Chulalongkorn University, Bangkok 10330, Thailand; pithi.c@chula.ac.th; 7Department of Pharmacology and Physiology, Faculty of Pharmaceutical Sciences, Chulalongkorn University, Bangkok 10330, Thailand; 8Department of Food and Pharmaceutical Chemistry, Faculty of Pharmaceutical Sciences, Chulalongkorn University, Bangkok 10330, Thailand; pornchai.r@chula.ac.th; 9Center of Excellence in Natural Products for Ageing and Chronic Diseases, Faculty of Pharmaceutical Sciences, Chulalongkorn University, Bangkok 10330, Thailand

**Keywords:** *Cymbidium ensifolium*, Orchidaceae, dihydrophenanthrene, dihydrophenanthrenequinone, anticancer

## Abstract

From the aerial parts of *Cymbidium ensifolium*, three new dihydrophenanthrene derivatives, namely, cymensifins A, B, and C (**1**–**3**) were isolated, together with two known compounds, cypripedin (**4**) and gigantol (**5**). Their structures were elucidated by analysis of their spectroscopic data. The anticancer potential against various types of human cancer cells, including lung, breast, and colon cancers as well as toxicity to normal dermal papilla cells were assessed via cell viability and nuclear staining assays. Despite lower cytotoxicity in lung cancer H460 cells, the higher % apoptosis and lower % cell viability were presented in breast cancer MCF7 and colon cancer CaCo_2_ cells treated with 50 µM cymensifin A (**1**) for 24 h compared with the treatment of 50 µM cisplatin, an available chemotherapeutic drug. Intriguingly, the half-maximum inhibitory concentration (IC_50_) of cymensifin A in dermal papilla cells at >200 µM suggested its selective anticancer activity. The obtained information supports the further development of a dihydrophenanthrene derivative from *C. ensifolium* as an effective chemotherapy with a high safety profile for the treatment of various cancers.

## 1. Introduction

*Cymbidium* spp. (Orchidaceae) comprises about fifty species widely dispersed across tropical and temperate Asia, especially in Thailand, China, and Nepal. Their beautiful flowers and structural appearance have made them more known for ornamental purposes rather than medicinal potentials. However, various parts of *Cymbidium* spp. are used in folkloric medicine to treat/manage many diseases. The leaves, roots, and pseudobulb of some *Cymbidium* spp. are employed in Nepali folkloric medicine for the treatment of boils, fever, paralysis, chronic illnesses, otitis, and as a tonic on rift bones [1]. Some species in this genus have been used as traditional Thai medicine. For example, the leaves of *C. aloifolium* and *C. findlaysonianum* have been used to treat otitis media [2]. *Cymbidium* plants are rich sources of phenanthrenes and bibenzyls, some of which exhibit several pharmacological activities such as cytotoxic, antimicrobial, free radical scavenging, and anti-inflammatory activities [3,4,5]. 

*Cymbidium ensifolium* (L.) Sw., known as “nang kham” or “chulan” in Thai [6], is an elegantly shaped orchid with beautiful aromatic flowers. Its flowers are white blended with yellow and red longitudinal lines along with the sepals and petals (Figure 1). It is distributed in north and northeastern Thailand [7]. The roots of this plant have been used as traditional Thai medicine to alleviate liver dysfunction and nephropathy [2,8]. In our continuing studies on bioactive compounds from orchids [9,10], we have explored the chemical constituents from the aerial parts of *C. ensifolium* with no previous reports on the chemical investigation. In the present study, a methanolic extract prepared from the aerial parts of this plant exhibited a significant cytotoxic effect against various cancer cells. This prompted us to investigate the plant to identify the cytotoxic compounds. 

## 2. Results and Discussion 

### 2.1. Structure Determination 

The phytochemical study of a methanolic extract from the leaves of *Cymbidium ensifolium* resulted in the purification of three novel dihydrophenanthrene derivatives (**1**–**3**) (Appendix A), along with two known compounds cypripedin (**4**) [11] and gigantol (**5**) [12] (Figure 2). The structures of the new compounds were elucidated through extensive spectroscopic data.

Compound **1** was purified as a red amorphous solid. The molecular formula C_16_H_14_O_5_ was analyzed from its [M−H]^−^ at 285.0770 (calcd. for C_16_H_13_O_5_, 285.0763) in the HR–ESI–MS. The IR spectrum displayed absorption bands for hydroxyl (3432 cm^−1^), aromatic (2924, 1638 cm^−1^), and ketone (1733 cm^−1^) groups. The UV absorptions at 255, 330, and 491 nm were indicative of a dihydrophenanthrenequinone nucleus [13]. The chemical structure was supported by the presence of two carbonyl carbons at δ 181.9 (C-1) and 188.0 (C-4). A 9,10-dihydro partial structure of **1** was confirmed by the signals of two methylene carbons at δ 20.0 (C-10) and 20.1 (C-9), which showed HSQC correlations to the protons at δ 2.59 (2H, dd, *J* = 8.4, 7.2 Hz, H_2_-10) and 2.80 (2H, m, H_2_-9), respectively. The ^1^H NMR spectrum of **1** also exhibited a singlet olefinic proton signal at δ 5.99 (1H, s, H-3), two doublet proton signals at δ 6.90 (1H, d, *J* = 8.8 Hz, H-6) and 7.63 (1H, d, *J* = 8.8 Hz, H-5), a singlet hydroxyl signal at δ 7.71 (HO-8) and signals for two methoxyl groups at δ 3.91 (3H, s, MeO-7) and 3.86 (3H, s, MeO-2) (Table 1). The quinone structure of ring A was deduced from the HMBC correlations of C-1 (δ 181.9) with H_2_-10 and H-3 (Figure 3). The first methoxyl group (δ 3.86) was located on ring A at C-2 based on its NOESY interaction with H-3. For ring B, the doublet proton signal at δ 7.63 was assigned as H-5 according to the HMBC correlations of C-4a (δ 137.3) with H-3 and H-5. The analysis of the ^1^H-^1^H COSY spectrum was supported by an *ortho*-coupled correlation between H-5 and H-6. The presence of a hydroxyl group at C-8 (δ 143.4) on ring B was supported by the HMBC correlation of C-8a (δ 125.6) with H-5, H_2_-10, and HO-8 and the NOESY crosspeak between HO-8 and H_2_-9. The second methoxyl group (δ 3.91) was substituted at C-7 (δ 150.0) according to its NOESY correlation with H-6 (Figure 3). Based on the above spectral data, **1** was characterized as 8-hydroxy-2,7-dimethoxy-9,10-dihydrophenanthrene-1,4-dione and named cymensifin A.

Compound **2**, a red amorphous solid, showed a similar molecular formula with **1** as C_16_H_14_O_5_ based on its [M−H]^−^ at 285.0764 (calcd. for C_16_H_13_O_5_, 285.0763) in the HR–ESI–MS. The UV absorptions and IR bands of **2** are similar with **1**, suggestive of a dihydrophenanthrenequinone skeleton. This was confirmed based on two carbonyl carbons at δ 181.8 (C-1) and 188.0 (C-4), and two methylene carbons at δ 20.5 (C-10) and 20.8 (C-9). The ^1^H NMR of **2** was similar to that of **1** by the presence of a singlet proton signal at δ 5.99 (1H, s, H-3), two aromatic doublet proton signals at δ 6.83 (1H, d, *J* = 8.8 Hz, H-6) and 7.75 (1H, d, *J* = 8.8 Hz, H-5), a singlet hydroxyl signal at δ 8.59 (HO-7), two methylene proton signal at δ 2.59 (2H, dd, *J* = 8.4, 7.6 Hz, H_2_-10) and 2.78 (2H, m, H_2_-9), and signals for two methoxyls at δ 3.76 (3H, s, MeO-8) and 3.86 (3H, s, MeO-2) (Table 1). On ring A of **2**, a methoxyl group (δ 3.86) was placed at C-2 according to its NOESY interaction with H-3. On ring B, the position of the methoxy and hydroxyl groups of **2** was alternated when compared with **1**. The second methyl group (δ 3.76) of **2** was located at C-8, as evidenced by the HMBC correlations of C-8 (δ 145.3) with MeO-8, H-6, H_2_-9, and HO-7. The HMBC correlations of C-7 (δ 153.1) with H-5 and HO-7 and the NOESY correlation between H-6 and HO-7 support the placement of the hydroxyl group at C-7 (Figure 3). Based on the above spectral evidence, 2 was identified as 7-hydroxy-2,8-dimethoxy-9,10-dihydrophenanthrene-1,4-dione and named cymensifin B.

Compound **3**, as a brown amorphous solid, gave a [M−H]^−^ at 301.1074 (calcd. for C_17_H_17_O_5_, 301.1076) in the HR –ESI–MS, suggesting a molecular formula C_17_H_18_O_5_. The IR spectrum demonstrated the exhibition of hydroxyl (3367 cm^−1^), aromatic ring (2924, 1606 cm^−1^), and methylene (1461 cm^−1^) functionalities. The UV absorptions at 279 and 312 nm and the ^13^C NMR signals of two methylene carbons at δ 21.8 (C-10) and 23.1 (C-9), which exhibited HSQC correlation with the ^1^H NMR signal at δ 2.71 (4H, br s, H_2_-9, H_2_-10), indicate a dihydrophenanthrene structure [14]. The ^1^H NMR displayed three aromatic proton signals at δ 6.52-7.70, resonances for three methoxy groups at δ 3.69 (3H, s, MeO-1), 3.79 (3H, s, MeO-4), and 3.85 (3H, s, MeO-7), and signals for two hydroxyl groups at δ 7.36 (1H, s, HO-8) and 8.16 (1H, s, HO-2) (Table 1). For ring A, the assignments of H-3 (δ 6.52, 1H, s), MeO-1, and HO-2 are in accord with the HMBC correlations of C-1 (δ 139.3) with H-3, H_2_-10, MeO-1, and HO-2 (Figure 3). A NOESY interaction of the second methoxy group at δ 3.79 with H-3 placed this methoxy group at C-4. Compound **3** exhibited structural similarity in ring B to that of **1** by the presence of two doublet proton signals of H-5 and H-6 at δ 7.70 (1H, d, *J* = 8.8 Hz) and 6.78 (1H, d, *J* = 8.8 Hz), respectively. A methoxy group at C-7 was supported by its NOESY interaction with H-6. A hydroxyl group at δ 7.36 was located at C-8 as evidenced by the HMBC correlations of C-8a (δ 124.8) with H-5, H_2_-10, and HO-8 (Figure 3). Compound **3** was assigned as 2,8-dihydroxy-1,4,7-trimethoxydihydrophananthrene and given the trivial name cymensifin C.

### 2.2. Cytotoxic Effects against Various Cancer Cells 

Because cytotoxic effects of cypripedin (**4**) and gigantol (**5**) were previously demonstrated in lung cancer cells [11,15], compounds **1**–**3** were evaluated for their anticancer potential against various human cancers, including lung, breast, and colon cancer cells. Cisplatin, a recommended chemotherapeutic agent [16], which at 50 µM caused the 50% reduction of viability in lung cancer cells, was selected for a positive control. The preliminary investigation via MTT assay showed that a culture with methanolic extract from *C. ensifolium* at 50 µg/mL for 24 h significantly diminished viability in lung cancer H460 and breast cancer MCF7 cells, but not in colon cancer CaCo_2_ cells when compared with control cells which were treated with the solvent vehicle 0.5% DMSO (Table 2). It should be noted that treatment with a lower concentration (10 μg/mL) of the methanolic extract did not obviously decrease cell viability in all cancer cells (data not shown). Among the three new dihydrophenanthrene derivatives, 24 h treatment with compound **1** (50 µM) showed the highest anticancer effect against H460, MCF7, and CaCo_2_ cells, as evidenced with the lowest % cell viability assessed via an MTT assay. When compared with cisplatin (Sigma-Aldrich Chemical, St. Louis, MO, USA), the higher anticancer potency of compound **1** was indicated with the lower % viability in MCF7 and CaCo_2_ cells. Additionally, Figure 4a–c illustrate the dose-response relationship of three new dihydrophenanthrene derivatives in H460, MCF7, and CaCo_2_ cells, respectively. In all cancer cells, compound **1** and cisplatin demonstrated anticancer activity in a concentration dependence. When compared with cisplatin treatment at the same concentration, the significantly lower % cell viability was presented in CaCo_2_ cells cultured with 50–200 μM of compound **1**. 

All data were presented as means ± standard error of the mean (SEM) from three independent experiments.

These were corresponded with apoptosis cell death detected via costaining with Hoechst 33342 and propidium iodide. Figure 5a–c respectively demonstrates apoptosis cells presenting with bright-blue fluorescence of condensed DNA or fragmented nuclei stained with Hoechst 33342 in H460, MCF7, and CaCo_2_ cells, which were cultured with either compound **1** (50 µM) or cisplatin (50 µM) for 24 h. Notably, there were no necrosis stained with the red fluorescence of propidium iodide in all treated cells. When compared with cisplatin treatment, the culture with compound **1** significantly increased % apoptosis in breast cancer MCF7 and colon cancer CaCo_2_ cells, although there was lower % apoptosis in lung cancer H460 cells (Figure 6a–c). 

It is the fact that the purity of an isolated natural compound critically influences its bioactivity [17]. Composed of known anticancer compounds, cypripedin (**4**) and gigantol (**5**), as well as compound **1**, a new dihydrophenanthrenequinone with a higher potency anticancer effect, the methanolic extract from *C. ensifolium* exhibited potent cytotoxicity against lung cancer and breast cancer cells (Table 2 and Figure 5a,b). Despite no toxicity presented in *C. ensifolium* extract-treated CaCo_2_ cells (Table 2 and Figure 5c), the dramatic reduction of % cell viability (Figure 4c) and accumulated apoptosis indicated in CaCo_2_ cells (Figure 5c and Figure 6c) incubated with compound **1** suggested specific cytotoxicity of compound **1** against colon cancer cells. 

As the most affected normal cells, human dermal papilla cells (DPCs) are chosen for evaluating the safety profile of potential anticancer compounds [18,19]. Intriguingly, DPCs cultured with 50 µM of three new dihydrophenanthrene derivatives (**1**–**3**) exhibited higher % cell viability (Table 2 and Figure 4d) and less apoptosis cells (Figure 5d and Figure 6d) compared with the treatment of cisplatin. Due to the highest anticancer potential among three new dihydrophenanthrene derivatives, the half-maximum inhibitory concentration (IC_50_) and selectivity index (S.I.) of compound **1** were calculated. Table 3 indicates that when compared with cisplatin, compound **1** possessed lower IC_50_ values in MCF7 (93.04 ± 0.86 µM) and CaCo_2_ cells (55.14 ± 3.08 µM), but it accounted for higher values of IC_50_ in H460 (66.71 ± 6.62 µM) and DPCs cells (>200 µM). It has been reported that the selectivity index of an anticancer agent, which selectively causes toxicity to cancer cells, should be more than 1 [20]. The higher selectivity index of compound **1** was indicated in MCF7 (>2.15) and CaCo_2_ cells (>3.62) compared with cisplatin treatment (MCF7: <0.59, CaCo_2_: <0.59). Although the selectivity index of compound **1** (>3.00) was lower than cisplatin (5.48 ± 0.25) in lung cancer cells, both values are more 1. These data clearly demonstrate the selective anticancer activity of compound **1** against various cancer cells.

Taken together, these results strongly suggest the potent anticancer effect against various types of cancer with the high safety profile of compound **1** isolated from the *C. ensifolium* extract. Nevertheless, the underlying mechanisms involved in the apoptosis inducing effect of compound **1** should be further investigated to clarify the therapeutic target.

## 3. Materials and Methods 

### 3.1. General Experimental Procedures 

UV spectra were measured on a Milton Roy Spectronic 300 Array spectrophotometer (Rochester, Monroe, NY, USA). IR spectra were recorded on a Perkin-Elmer FT-IR 1760X spectrophotometer (Boston, MA, USA). Mass spectra were obtained on a Bruker micro TOF mass spectrometer (ESI-MS) (Billerica, MA, USA). NMR spectra were recorded on a Bruker Avance Neo 400 MHz NMR spectrometer (Billerica, MA, USA). Vacuum liquid column chromatography (VLC) and column chromatography (CC) were performed on silica gel 60 (Merck, Kieselgel 60, 70–320 µm) and silica gel 60 (Merck, Kieselgel 60, 230–400 µm) (Darmstadt, Germany).

### 3.2. Plant Material

*Cymbidium ensifolium* was purchased from Chatuchak market, Bangkok, in September 2020. Plant identification was performed by Mr. Yanyong Punpreuk, Department of Agriculture, Bangkok, Thailand. A voucher specimen (BS-CE-092563) has been deposited at the Department of Pharmacognosy and Pharmaceutical Botany, Faculty of Pharmaceutical Sciences, Chulalongkorn University.

### 3.3. Extraction and Isolation

The air-dried aerial parts of *Cymbidium ensifolium* (798 g) were cut, pulverized, and extracted with MeOH (3 × 6 L) at room temperature. The resultant organic solvent was evaporated under reduced pressure to give a dried mass (97.68 g). The methanolic extract was fractionated by vacuum liquid chromatography on silica gel (hexane–ethyl acetate, gradient) to give five fractions (A–E). Fraction C (23 g) was separated by column chromatography (CC, silica gel hexane–ethyl acetate, gradient) to yield seven fractions (C1–C7). Fraction C3 (46.8 mg) was further separated by CC (silica gel, hexane–dichloromethane, gradient) and then purified on silica gel (hexane–dichloromethane–methanol, gradient) to afford **1** (3 mg). Fractions C5 (32.6 mg) and C7 (25.1 mg) were separated on a hexane–dichloromethane gradient and thereafter, on silica gel (hexane–dichloromethane–methanol gradient) to yield cypripedin (**4**) (3.2 mg) and **2** (4 mg), respectively. Fraction D (16 g) was fractionated by CC (silica gel, hexane–ethyl acetate, gradient) to yield 12 fractions (D1–D12). Fraction D1 (76.3 mg) was further separated on silica gel (hexane–dichloromethane–methanol, gradient) to furnish **3** (1.8 mg). Gigantol (**5**) (2.1 mg) was obtained from fraction D7 (8.0 mg) after purification on Sephadex LH-20 (acetone).

Cymensifin A (**1**): Red amorphous solid; UV (MeOH) λ_max_ (log ε): 255 (4.86), 330 (2.55), 491 (1.28) nm; IR (film) ν_max_: 3432, 2924, 2854, 1733, 1638, 1461, 1271 cm^−1^; HR–ESI–MS: [M−H]^−^ at *m*/*z* 285.0770 (calcd. for 285.0763, C_16_H_13_O_5_).

Cymensifin B (**2**): Red amorphous solid; UV (MeOH) λ_max_ (log ε): 254 (4.32), 333 (2.13), 493 (1.01) nm; IR (film) ν_max_: 3370, 2924, 2854, 1732, 1642, 1460, 1246 cm^−1^; HR–ESI–MS: [M−H]^−^ at *m*/*z* 285.0764 (calcd. for 285.0763, C_16_H_13_O_5_).

Cymensifin C (**3**): Brown amorphous solid; UV (MeOH) λ_max_ (log ε): 279 (3.74), 312 (2.48) nm; IR (film) ν_max_: 3367, 2924, 2854, 1606, 1510, 1461, 1246 cm^−1^; HR–ESI–MS: [M−H]^−^ at *m*/*z* 301.1074 (calcd. for 301.1076, C_17_H_17_O_5_).

### 3.4. Cell Culture

Human lung cancer H460, breast cancer MCF7, and colon cancer CaCo_2_ cells were obtained from The American Type Culture Collection (ATCC, Manassas, VA, USA). Lung cancer H460 cells were cultured in the Roswell Park Memorial Institute (RPMI; Gibco, Gaithersburg, MA, USA) medium whereas human MCF7 and CaCo_2_ cells were maintained in Dulbecco’s modified eagle medium (DMEM; Gibco, Gaithersburg, MA, USA). Human dermal papilla cells (DPCs) purchased from Applied Biological Materials Inc. (Richmond, BC, Canada) were cultured in a Prigrow III medium (Applied Biological Materials Inc., Richmond, BC, Canada). All culture mediums were supplemented with 2 mM l-glutamine, 10% FBS (fetal bovine serum) and 100 units/mL penicillin/streptomycin. The cells were maintained in at 37 °C with 5% CO_2_ until 70–80% confluence before using in further experiments.

### 3.5. Cell Viability Assay

After 24 h of indicated treatment, cells seeded at a density of 1 × 10^4^ cells/well in a 96-well plate was further incubated with 0.4 mg/mL of MTT (3-(4,5-dimethylthiazol-2-yl)-2,5-diphenyltetrazolium bromide; Sigma-Aldrich Chemical, St. Louis, MO, USA) for 3 h at 37 °C and kept from light. Then, the formed formazan crystal was solubilized in a DMSO prior measurement of optical density (OD) at 570 nm by using a microplate reader (Anthros, Durham, NC, USA). The OD ratio of treated to control cells, which were incubated with 0.5% DMSO as a solvent vehicle, was calculated and presented as percent cell viability [19,21].

Furthermore, the half-maximum inhibitory concentration (IC_50_) was calculated and used for determination of selective index in each cancer cell. The selectivity index was represented from the ratio between the IC_50_ value in dermal papilla cells and the IC_50_ value in cancer cells [20]. 

### 3.6. Detection of Mode of Cell Death 

Apoptosis and necrosis cell death were evaluated by nuclear staining assay. The treated cells were costained with 0.02 µg/mL Hoechst 33342 (Sigma-Aldrich Chemical, St. Louis, MO, USA) and 0.01 µg/mL propidium iodide (Sigma-Aldrich Chemical, St. Louis, MO, USA) at 37 °C for 30 min. The mode of cell death was visualized under a fluorescence microscope (Olympus IX51 with DP70, Olympus, Tokyo, Japan). Apoptosis cells were characterized with a bright-blue fluorescence of Hoechst 33342 which stained fragmented DNA and condensed nuclei. Meanwhile, necrosis cells were distinguished by red fluorescence of propidium iodide. The percentage of apoptosis was represented from the ratio between the number of apoptosis cells to the total cell number [19,22]. 

### 3.7. Statistical Analysis

The statistical analysis was performed via SPSS version 22 (IBM Corp., Armonk, NY, USA) with one-way analysis of variance (ANOVA) followed by Tukey’s post hoc test. Any *p*-value under 0.05 was considered as a statistical significance.

## 4. Conclusions

In this study, three novel dihydrophenanthrenes derivatives, cymensifins A–C (**1**–**3**), together with two known compounds, cypripedin (**4**) and gigantol (**5**), were isolated from the aerial parts of *C. ensifolium*. The structures of the new compounds were elucidated by spectroscopic analysis. The three new compounds from this plant were evaluated for their cytotoxicity on human lung cancer H460, breast cancer MCF7, and colon cancer CaCo_2_ cells. Cymensifin A (**1**) showed a promising anticancer effect against various cancer cells with higher safety profiles compared with cisplatin, an available chemotherapy. To the best of our knowledge, this research is the first record on the chemical constituents and biological activity of this plant.

## Figures and Tables

**Figure 1 molecules-27-02222-f001:**
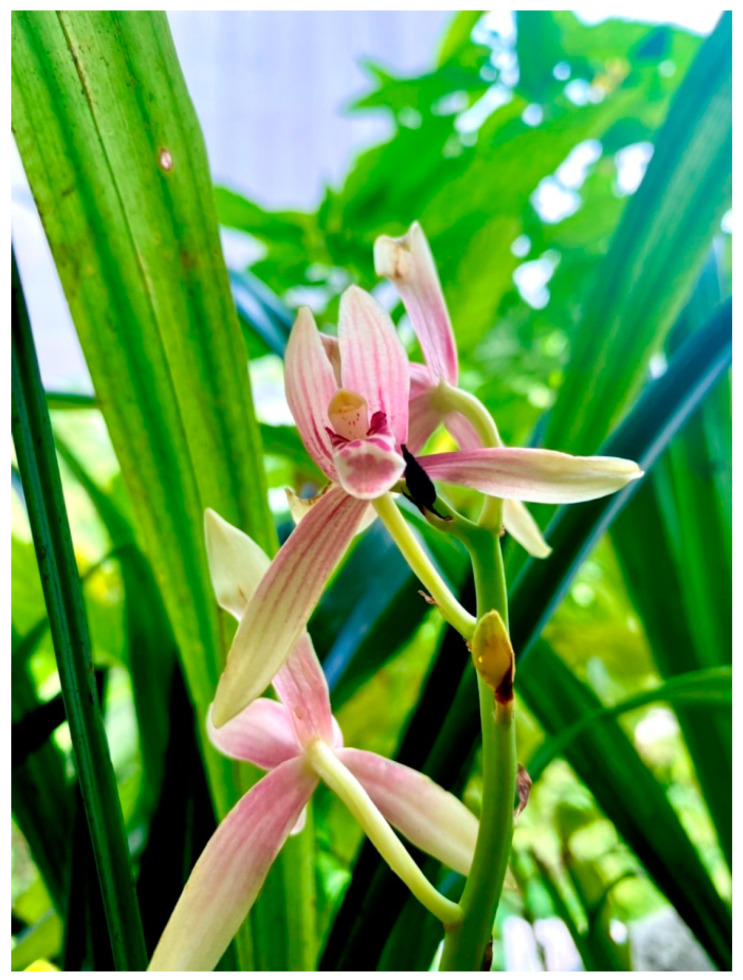
*Cymbidium ensifolium* (L.) Sw.

**Figure 2 molecules-27-02222-f002:**
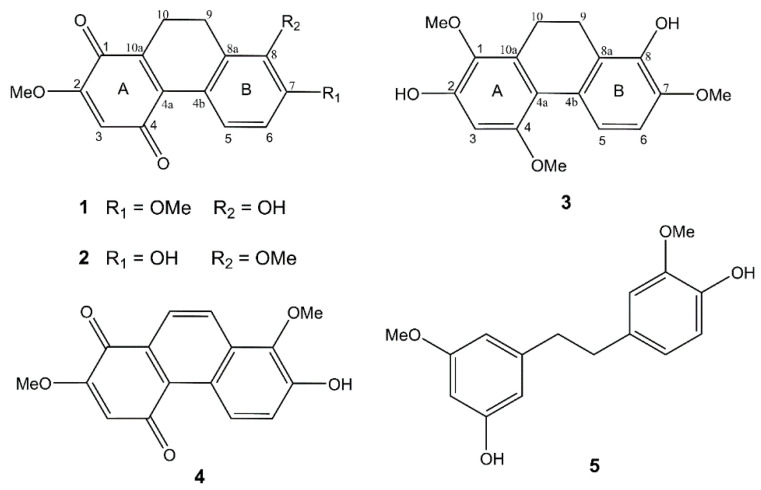
Chemical structures of compounds **1**–**5** isolated from *Cymbidium ensifolium*.

**Figure 3 molecules-27-02222-f003:**
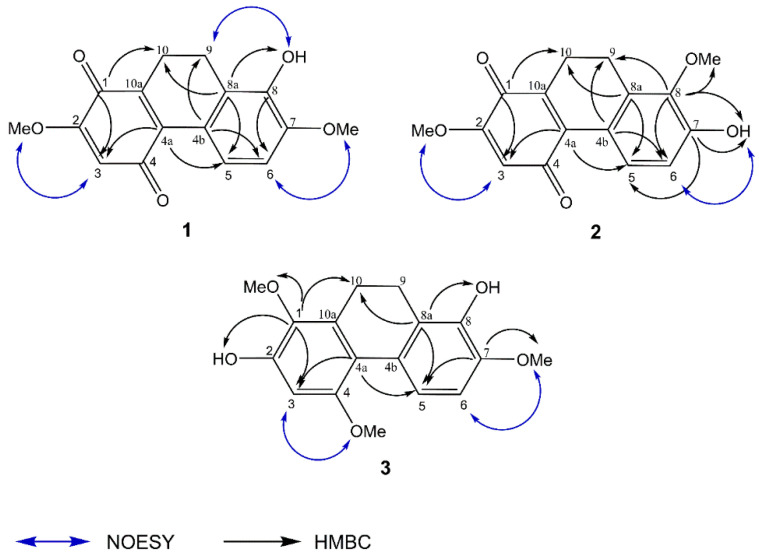
Selected HMBC and NOESY correlations of **1**, **2**, and **3**.

**Figure 4 molecules-27-02222-f004:**
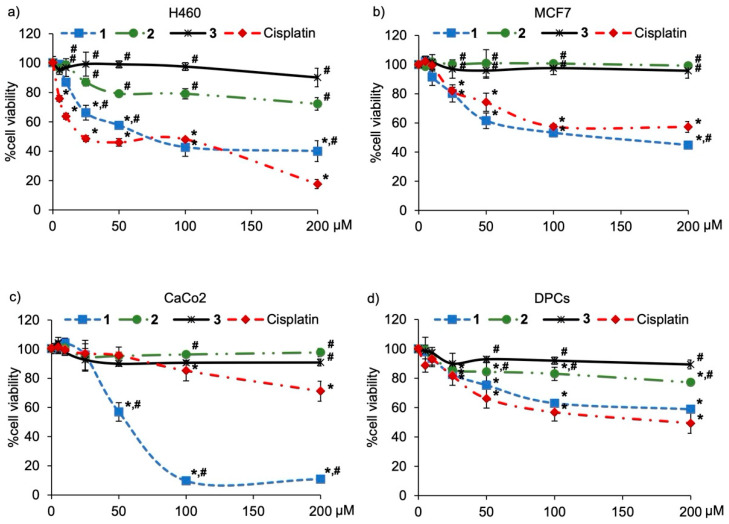
The relationship between cytotoxicity and the concentration of dihydrophenanthrene derivatives (**1**–**3**) from *Cymbidium ensifolium* in (**a**) lung cancer H460, (**b**) breast cancer MCF7, (**c**) colon cancer CaCo_2_, and (**d**) dermal papilla cells (DPCs). Cisplatin, a recommended anticancer drug, was used as a positive control. The data were presented as means ± standard error of the mean (SEM) from three independent experiments. * *p* < 0.05 compared with control cells treated with 0.5% DMSO as the solvent vehicle, ^#^
*p* < 0.05 compared with treatment of cisplatin at the same concentration.

**Figure 5 molecules-27-02222-f005:**
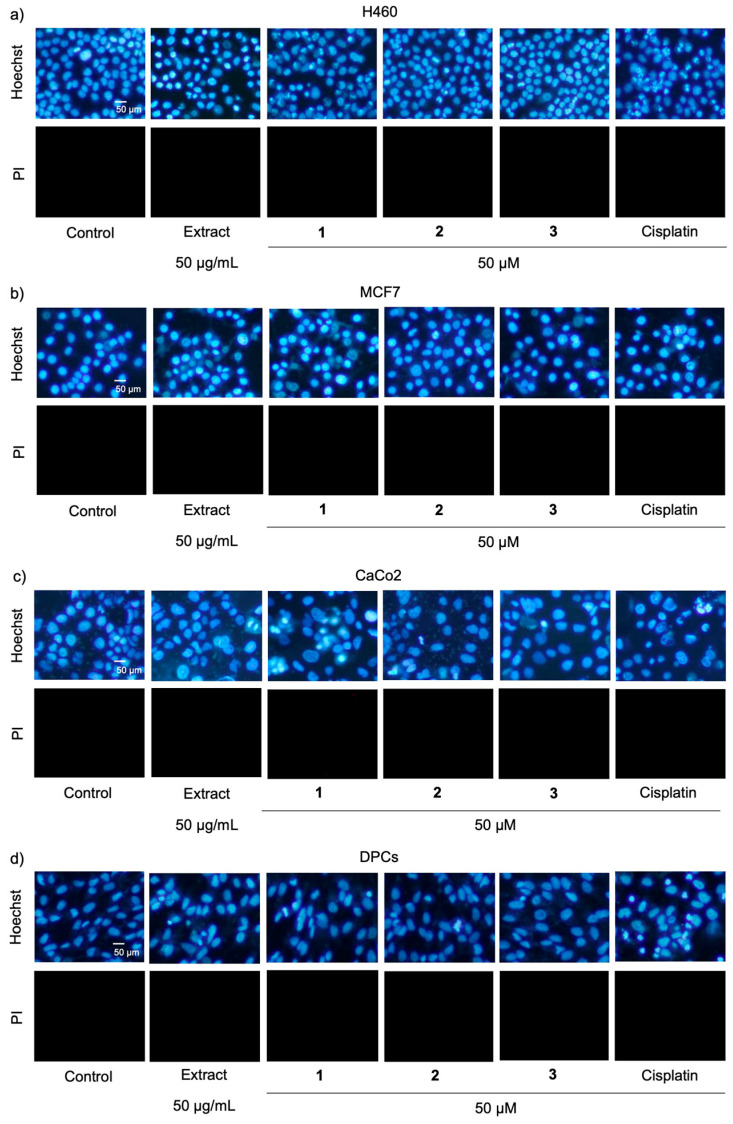
A nuclear staining assay demonstrated apoptosis presented with bright-blue fluorescence of Hoechst 33342 in (**a**) lung cancer H460, (**b**) breast cancer MCF7, (**c**) colon cancer CaCo_2_, and (**d**) dermal papilla cells (DPCs) cultured with methanolic extract and isolated compounds **1**–**3** from *Cymbidium ensifolium* for 24 h. Cisplatin, a recommended anticancer drug, was used as a positive control. Notably, there were no detected necrosis cells stained with red fluorescence of propidium iodide (PI) in all treated cells.

**Figure 6 molecules-27-02222-f006:**
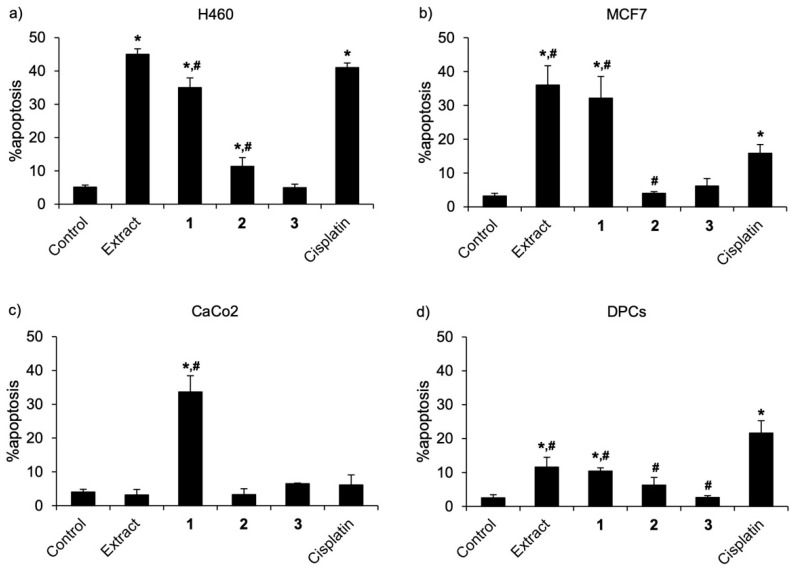
The percent (%) apoptosis calculated from nuclear staining assay in (**a**) lung cancer H460, (**b**) breast cancer MCF7, (**c**) colon cancer CaCo_2_, and (**d**) dermal papilla cells (DPCs) cultured with methanolic extract and isolated compounds **1**–**3** from *Cymbidium ensifolium* for 24 h. Cisplatin, a recommended anticancer drug, was used as a positive control. The data were presented as means ± standard error of the mean (SEM) from three independent experiments. * *p* < 0.05 compared with control cells treated with 0.5% DMSO as a solvent vehicle, ^#^
*p* < 0.05 compared with the cisplatin treatment group.

**Table 1 molecules-27-02222-t001:** ^1^H (400 MHz) and ^13^C-NMR (100 MHz) spectral data of **1**–**3** in acetone-*d*_6_.

	1		2		3	
Position	δ_H_ (Multiplicity, *J* in Hz)	δ_C_	δ_H_ (Multiplicity, *J* in Hz)	δ_C_	δ_H_ (Multiplicity, *J* in Hz)	δ_C_
1	-	181.9	-	181.8	-	139.3
2	-	159.5	-	159.5	-	149.8
3	5.99 (s)	108.6	5.99 (s)	108.5	6.52 (s)	99.9
4	-	188.0	-	188.0	-	154.9
4a	-	137.3	-	137.2	-	116.7
4b	-	123.8	-	123.0	-	127.6
5	7.63 (d, *J* = 8.8 Hz)	123.8	7.75 (d, *J* = 8.8 Hz)	128.4	7.70 (d, *J* = 8.8 Hz)	120.4
6	6.90 (d, *J* = 8.8 Hz)	109.3	6.83 (d, *J* = 8.8 Hz)	114.9	6.78 (d, *J* = 8.8 Hz)	109.0
7	-	150.0	-	153.1	-	146.3
8	-	143.4	-	145.3	-	142.9
8a	-	125.6	-	134.0	-	124.8
9	2.80 (m)	20.1	2.78 (m)	20.8	2.71 (br s)	23.1
10	2.59 (dd, *J* = 8.4, 7.2 Hz)	20.0	2.59 (dd, *J* = 8.4, 7.6 Hz)	20.5	2.71 (br s)	21.8
10a	-	137.9	-	137.4	-	133.6
MeO-1	-	-	-	-	3.69 (s)	61.1
MeO-2	3.86 (s)	56.6	3.86 (s)	56.6	-	-
MeO-4	-	-	-	-	3.79 (s)	56.1
MeO-7	3.91 (s)	56.3	-	-	3.85 (s)	56.3
MeO-8	-	-	3.76 (s)	61.0	-	-
HO-2	-	-	-	-	8.16 (s)	-
HO-7	-	-	8.59 (s)	-	-	-
HO-8	7.71 (s)	-	-	-	7.36 (s)	-

**Table 2 molecules-27-02222-t002:** Cell viability percentage in various cancer and normal cells cultured with 50 µM of new dihydrophenanthrene derivatives (**1**–**3**) from *Cymbidium ensifolium* for 24 h.

Cell Type	% Cell Viability				
	Extract (50 µg/mL)	1 (50 µM)	2 (50 µM)	3 (50 µM)	Cisplatin (50 µM)
H460	43.49 ± 6.42 *	57.65 ± 1.39 *^,#^	79.48 ± 1.11 ^#^	99.12 ± 2.91 ^#^	46.02 ± 2.07 *
MCF7	52.76 ± 2.09 *	61.51 ± 5.46 *	100.45 ± 9.85 ^#^	96.45 ± 5.11 ^#^	74.05 ± 5.69 *
CaCo_2_	95.37 ± 2.21	56.92 ± 6.35 *^,#^	95.26 ± 5.05	90.96 ± 1.44	90.20 ± 4.27
DPCs	73.41 ± 4.72 *	75.66 ± 1.28 *	84.39 ± 5.78 *^,#^	92.68 ± 2.22 ^#^	66.18 ± 6.44 *

* *p* < 0.05 compared with control cells treated with 0.5% DMSO as the solvent vehicle, ^#^
*p* < 0.05 compared with the cisplatin treatment group.

**Table 3 molecules-27-02222-t003:** The half-maximum inhibitory concentration (IC_50_) and selectivity index (S.I.) of compound **1** from *Cymbidium ensifolium* and cisplatin.

Cell Type	1		Cisplatin	
	IC_50_ (µM)	S.I.	IC_50_ (µM)	S.I.
H460	66.71 ± 6.62	>3.00	21.57 ± 4.13	5.48 ± 0.25
MCF7	93.04 ± 0.86	>2.15	>200	<0.59
CaCo_2_	55.14 ± 3.08	>3.62	>200	<0.59
DPCs	>200	N/A	117.63 ± 17.19	N/A

N/A: not applicable. All data were presented as means ± standard error of the mean (SEM) from three independent experiments.

## Data Availability

All data presented in this study are available in the article.

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
