# Peer review of "Three New Dihydrophenanthrene Derivatives from *Cymbidium ensifolium* and Their Cytotoxicity against Cancer Cells"

_molecules, 2022, doi:10.3390/molecules27072222_

Round 1
Reviewer 1 Report
The authors describe the isolation of novel new phenanthrene compounds with potential anticancer activity. Although the structure of the manuscript and results are systematically described, minor revisions are required
- several English mistakes can be detected in terms of spelling and sentence structuring. I would suggest revision by an English speaker
- the conclusion at the end of the abstract is very weak and need to be changed
- what exactly is the aerial parts used in the study? how is the author sure that it is not contaminated by other parts of the same plants or other plants. did the plants purchase fresh or dry? How long is the drying process and how? what is the temp employed for drying?
- cytotoxic assay please add details according to published standard procedure
- the results of cytotoxicity is better to be presented in a graph to represent all statistics and so can clearly find the significance.
- the choice of fractions employed for compound isolation is not clear why or depend on what?
- I would expect a "fractionation-based cytotoxic assay" rather than cytotoxicity?
- The difference in nuclear staining assay is not clear, I would suggest also presented as a graph as well which can be measured based on density?
- Please mention the control used in the cytotoxicity and mechanisms
Author Response
Reviewer #1
The authors describe the isolation of novel new phenanthrene compounds with potential anticancer activity. Although the structure of the manuscript and results are systematically described, minor revisions are required
1. several English mistakes can be detected in terms of spelling and sentence structuring. I would suggest revision by an English speaker
Response: The spelling and grammar mistakes had been corrected. Moreover, the revised manuscript had been proofed by an English native speaker.
2. the conclusion at the end of the abstract is very weak and need to be changed
Response: The end of abstract had been corrected as suggestion by reviewer to emphasize the potential anticancer activity of new phenanthrene compounds and presented in Abstract of the revised manuscript as “Despite lower cytotoxicity in lung cancer H460 cells, the higher %apoptosis and lower %cell viability were presented in breast cancer MCF7 and colon cancer CaCo2 cells treated with 50 µM cymensifin A (1) for 24 h compared with the treatment of 50 µM cisplatin, an available chemo-therapeutic drug. Intriguingly, the half-maximum inhibitory concentration (IC50) of cymensifin A in dermal papilla cells at > 200 µM suggested its selective anticancer activity. The obtained information supports the further development of dihydrophenanthrene derivative from C. ensifolium as an effective chemotherapy with high safety profile for treatment of various cancers.”
3. what exactly is the aerial parts used in the study? how is the author sure that it is not contaminated by other parts of the same plants or other plants. did the plants purchase fresh or dry? How long is the drying process and how? what is the temp employed for drying?
Response: The aerial parts (stems and leaves) and the roots of this plant can be definitely separated. We got this plant in fresh plant. Adulterants were removed before chopping and drying. For drying process, hot air oven at 60 °C, 8 hr per day, until the drying process was complete.
4. cytotoxic assay please add details according to published standard procedure
Response: The detail cytotoxic assay and references of the standard and acceptable protocol had been added and presented in the Materials and Methods section under sub-topic of 3.5. Cell viability assay of the revised manuscript as “After 24 h of indicated treatment, cells seeded at a density of 1 × 104 cells/well in 96-well plate was further incubated with 0.4 mg/mL of MTT (3-(4,5-dimethylthiazol-2-yl)-2,5-diphenyltetrazolium bromide; Sigma-Aldrich Chemical, St. Louis, MO, USA) for 3 h at 37°C kept from light. Then, the formed formazan crystal was solubilized in DMSO prior measurement of optical density (OD) at 570 nm by using a microplate reader (Anthros, Durham, NC, USA). The OD ratio of treated to control cells, which were incubated with 0.5% DMSO as vehicle solvent was calculated and presented as percent cell viability [19,21].
Furthermore, the half-maximum inhibitory concentration (IC50) was calculated and used for determination of selective index in each cancer cells. The selectivity index was represented from the ratio between IC50 value in dermal papilla cells and IC50 value in cancer cells [20].”
Reference as presented in manuscript
- Khine, H.E.E.; Ecoy, G.A.U.; Roytrakul, S.; Phaonakrop, N.; Pornputtapong, N.; Prompetchara, E.; Chanvorachote, P.; Chaotham, C. Chemosensitizing activity of peptide from Lentinus squarrosulus (Mont.) on cisplatin-induced apoptosis in human lung cancer cells. Sci Rep. 2021, 11, 4060.
- Lica, J.J.; Wieczór, M.; Grabe, G.J.; Heldt, M.; Jancz, M.; Misiak, M.; Gucwa, K.; Brankiewicz, W.; Maciejewska, N.; Stupak, A.; Bagiński, M.; Rolka, K.; Hellmann, A.; Składanowski, A. Effective drug concentration and selectivity depends on fraction of primitive cells. Int J Mol Sci. 2021, 22, 4931.
- Marks, D.C.; Belov, L.; Davey, M.W.; Davey, R.A.; Kidman, A.D. The MTT cell viability assay for cytotoxicity testing in multidrug-resistant human leukemic cells. Leuk Res. 1992, 16, 1165-1173.
5. the results of cytotoxicity is better to be presented in a graph to represent all statistics and so can clearly find the significance.
Response: The dose-response relationship of each compound was presented in Figure 4 and presented in the Results and Discussion section under sub-topic of 2.2. Cytotoxic effects against various cancer cells of the revised manuscript as “Additionally, Figure 4a, b and c illustrate the dose-response relationship of three new dihydrophenanthrene derivatives in H460, MCF7 and CaCo2 cells, respectively. In all cancer cells, compound 1 and cisplatin demonstrated anticancer activity in a concentration dependence. It was worth to noted that when compared with cisplatin treatment at the same concentration, the significantly lower %cell viability was presented in CaCo2 cells cultured with 50–200 mM of compound 1.”
6. the choice of fractions employed for compound isolation is not clear why or depend on what?
Response: Fractions were selected for further isolation of pure compounds based on their TLC analysis.
7. I would expect a "fractionation-based cytotoxic assay" rather than cytotoxicity?
Response: Thank you for your comments but we did not perform fractionation-based cytotoxic assay.
8. The difference in nuclear staining assay is not clear, I would suggest also presented as a graph as well which can be measured based on density?
Response: The percentage of apoptosis cells was counted and represented as the ratio between number of apoptosis cells to total cell number in Figure 6 and presented in the Results and Discussion section under sub-topic of 2.2. Cytotoxic effects against various cancer cells of the revised manuscript as “When compared with cisplatin treatment, culture with compound 1 significantly increased %apoptosis in breast cancer MCF7 and colon cancer CaCo2 cells although there was the lower %apoptosis in lung cancer H460 cells (Figure 6a–c).”
9. Please mention the control used in the cytotoxicity and mechanisms
Response: The control group that was treated with 0.5% DMSO as vehicle solvent was mention in the revised manuscript.
Reviewer 2 Report
You cannot compare 50 ug/mL of an extract with 50 uM of a single compound. The dose of the extract is immensely higher.
Furthermore, it is an absurd failing not to have performed the assays with other concentrations of the compounds (at least 4).
The IC50 of cisplatin is known; but what is the IC50 of the isolated/derived compounds on the tumour lines evaluated? Dose dependent?
No selectivity calculation or Cell Cycle and DNA Content Analysis or Western Blotting.
In the current scenario, where thousands of molecules with antitumour activity are already known, a study that wants to present new molecules cannot do it only with a simple MTT. Much more information regarding these molecules - and especially how and where they act on tumour cells - must be substantially clarified.
The manuscript does not present any discussion. Everything written in section 2 is solely and exclusively description of the results.
Author Response
Reviewer #2
- You cannot compare 50 ug/mL of an extract with 50 uM of a single compound. The dose of the extract is immensely higher.
Response: Thank for valuable comments from reviewer. The cytotoxicity of extract against various cancer cells was initially evaluated at concentration of 10 and 50 μg/mL to screen the potential anticancer activity however only the extract at 50 μg/mL significantly reduced %viability in human lung and breast cancer cells. The explanation of this preliminary screening was presented in the Results and Discussion section under sub-topic of 2.2. Cytotoxic effects against various cancer cells of the revised manuscript as “The preliminary investigation via MTT assay showed that culture with methanolic extract from C. ensifolium at 50 mg/mL for 24 h significantly diminished viability in lung cancer H460 and breast cancer MCF7 cells but not in colon cancer CaCo2 cells when compared with control cells which were treated with vehicle solvent, 0.5% DMSO (Table 2). It should be noted that treatment with lower concentration (10 μg/mL) of the methanolic extract did not obviously decrease cell viability in all cancer cells (data not shown).”
- Furthermore, it is an absurd failing not to have performed the assays with other concentrations of the compounds (at least 4).
Response: The dose-response relationship of each compound was presented in Figure 4 and presented in the Results and Discussion section under sub-topic of 2.2. Cytotoxic effects against various cancer cells of the revised manuscript as “Additionally, Figure 4a, b and c illustrate the dose-response relationship of three new dihydrophenanthrene derivatives in H460, MCF7 and CaCo2 cells, respectively. In all cancer cells, compound 1 and cisplatin demonstrated anticancer activity in a concentration dependence. It was worth to noted that when compared with cisplatin treatment at the same concentration, the significantly lower %cell viability was presented in CaCo2 cells cultured with 50–200 mM of compound 1.”
- The IC50 of cisplatin is known; but what is the IC50 of the isolated/derived compounds on the tumour lines evaluated? Dose dependent?
Response: The IC50 of compound 1 that exhibits the most potency on anticancer activity among three new dihydrophenanthrene derivatives was presented in Table 3 and in the Results and Discussion section under sub-topic of 2.2. Cytotoxic effects against various cancer cells of the revised manuscript as “Due to the highest anticancer potential among three new dihydrophenanthrene derivatives, the half-maximum inhibitory concentration (IC50) and selectivity index (S.I.) of compound 1 were calculated. Table 3 indicates that when compare with cisplatin, compound 1 possessed lower IC50 values in MCF7 (93.04 ± 0.86 µM) and CaCo2 cells (55.14 ± 3.08 µM) but it accounted for higher values of IC50 in H460 (66.71 ± 6.62 µM) and DPCs cells (> 200 µM).”
- No selectivity calculation or Cell Cycle and DNA Content Analysis or Western Blotting.
In the current scenario, where thousands of molecules with antitumour activity are already known, a study that wants to present new molecules cannot do it only with a simple MTT. Much more information regarding these molecules - and especially how and where they act on tumour cells - must be substantially clarified.
Response: Due to the low amount of compounds 1-3, the anticancer effect evaluated in this study was a preliminary investigation. The investigation on related-underlying mechanism should be further investigated. This was explained in the Results and Discussion section under sub-topic of 2.2. Cytotoxic effects against various cancer cells of the revised manuscript as “Taken together, these results strongly suggest the potent anticancer effect against various types of cancer with high safety profile of compound 1 isolated from C. ensifolium extract. Nevertheless, the underlying mechanisms involving in apoptosis inducing effect of compound 1 should be further investigated to clarify the therapeutic target.”
Nevertheless, the selectivity index of compound 1 was that exhibits the most potency on anticancer activity among three new dihydrophenanthrene derivatives was presented in Table 3 and in the Results and Discussion section under sub-topic of 2.2. Cytotoxic effects against various cancer cells of the revised manuscript as “It has been reported that selectivity index of anticancer agent which selectively causes toxicity to cancer cells should be more than 1 [20]. The higher selectivity index of compound 1 was indicated in MCF7 (> 2.15) and CaCo2 cells (> 3.62) compared with cisplatin treatment (MCF7: < 0.59, CaCo2: < 0.59). Although the selectivity index of compound 1 (> 3.00) is lower than cisplatin (5.48 ± 0.25) in lung cancer cells, both values are more 1. These data clearly demonstrate the selective anticancer activity of compound 1 against various cancer cells.”
Reference as presented in manuscript
- Lica, J.J.; Wieczór, M.; Grabe, G.J.; Heldt, M.; Jancz, M.; Misiak, M.; Gucwa, K.; Brankiewicz, W.; Maciejewska, N.; Stupak, A.; Bagiński, M.; Rolka, K.; Hellmann, A.; Składanowski, A. Effective drug concentration and selectivity depends on fraction of primitive cells. Int J Mol Sci. 2021, 22, 4931.
5. The manuscript does not present any discussion. Everything written in section 2 is solely and exclusively description of the results.
Response: The Results and Discussion under sub-topic of 2.2. Cytotoxic effects against various cancer cells of the revised manuscript had been corrected as suggested by reviewer.
Reviewer 3 Report
The paper presents the isolation of three novel dihydrophenanthrenes derivatives (1-3) and two known (4-5) compounds from the aerial parts of Cymbidium ensifolium, and their anticancer potential against three types of human cancer cells. The structures of new compounds were elucidated through spectroscopic data. Compound 1 showed a promising anticancer effect against various cancer cells with higher safety profiles compared with cisplatin.
In my opinion the manuscript contains enough original and interesting material. It is written clearly and concisely. The experimental procedures are described comprehensively. The results are interesting.
Minor corrections:
It is worth calculating the selectivity indices.
Why was the cytotoxic activity of the extract tested at a concentration of 50 μg mL-1, while the cytotoxic activity of compounds 1-3 and cisplatin at a concentration of 50 μM?
Figure 4 should be placed below Table 2.
The authors did not include the 1H and 13C NMR spectra in the Supplementary material.
Author Response
Reviewer #3
The paper presents the isolation of three novel dihydrophenanthrenes derivatives (1-3) and two known (4-5) compounds from the aerial parts of Cymbidium ensifolium, and their anticancer potential against three types of human cancer cells. The structures of new compounds were elucidated through spectroscopic data. Compound 1 showed a promising anticancer effect against various cancer cells with higher safety profiles compared with cisplatin.
In my opinion the manuscript contains enough original and interesting material. It is written clearly and concisely. The experimental procedures are described comprehensively. The results are interesting.
Minor corrections:
- It is worth calculating the selectivity indices.
Response: The selectivity index of compound 1 was that exhibits the most potency on anticancer activity among three new dihydrophenanthrene derivatives was presented in Table 3 and in the Results and Discussion section under sub-topic of 2.2. Cytotoxic effects against various cancer cells of the revised manuscript as “It has been reported that selectivity index of anticancer agent which selectively causes toxicity to cancer cells should be more than 1 [20]. The higher selectivity index of compound 1 was indicated in MCF7 (> 2.15) and CaCo2 cells (> 3.62) compared with cisplatin treatment (MCF7: < 0.59, CaCo2: < 0.59). Although the selectivity index of compound 1 (> 3.00) is lower than cisplatin (5.48 ± 0.25) in lung cancer cells, both values are more 1. These data clearly demonstrate the selective anticancer activity of compound 1 against various cancer cells.”
Reference as presented in manuscript
- Lica, J.J.; Wieczór, M.; Grabe, G.J.; Heldt, M.; Jancz, M.; Misiak, M.; Gucwa, K.; Brankiewicz, W.; Maciejewska, N.; Stupak, A.; Bagiński, M.; Rolka, K.; Hellmann, A.; Składanowski, A. Effective drug concentration and selectivity depends on fraction of primitive cells. Int J Mol Sci. 2021, 22, 4931.
2. Why was the cytotoxic activity of the extract tested at a concentration of 50 μg mL-1, while the cytotoxic activity of compounds 1-3 and cisplatin at a concentration of 50 μM?
Response: The cytotoxicity against various cancer cells of extract was initially evaluated at concentration of 10 and 50 µg/mL to screen the potential anticancer activity however only the extract at 50 µg/mL significantly reduced %viability in human lung and breast cancer cells. The explanation of this preliminary screening was presented in the Results and Discussion section under sub-topic of 2.2. Cytotoxic effects against various cancer cells of the revised manuscript as “The preliminary investigation via MTT assay showed that culture with methanolic extract from C. ensifolium at 50 mg/mL for 24 h significantly diminished viability in lung cancer H460 and breast cancer MCF7 cells but not in colon cancer CaCo2 cells when compared with control cells, which were treated with vehicle solvent, 0.5% DMSO (Table 2). It should be noted that treatment with lower concentration (10 μg/mL) of the methanolic extract did not obviously decrease cell viability in all cancer cells (data not shown).”
To compare cytotoxicity with cisplatin, the IC50 of compound 1 that exhibits the most potency on anticancer activity among three new dihydrophenanthrene derivatives was presented in Table 3 and in the Results and Discussion section under sub-topic of 2.2. Cytotoxic effects against various cancer cells of the revised manuscript as “Due to the highest anticancer potential among three new dihydrophenanthrene derivatives, the half-maximum inhibitory concentration (IC50) and selectivity index (S.I.) of compound 1 were calculated. Table 3 indicates that when compare with cisplatin, compound 1 possessed lower IC50 values in MCF7 (93.04 ± 0.86 µM) and CaCo2 cells (55.14 ± 3.08 µM) but it accounted for higher values of IC50 in H460 (66.71 ± 6.62 µM) and DPCs cells (> 200 µM).”
- Figure 4 should be placed below Table 2.
Response: Due to adding of new results the sequence of Tables and figures was corrected as presented in the revised manuscript.
- The authors did not include the 1H and 13C NMR spectra in the Supplementary material.
Response: The 1H and 13C NMR spectra and 2D NMR data have been added in the Supplementary material.
Round 2
Reviewer 2 Report
The authors themselves acknowledge that the data presented are preliminary and that few studies were performed due to the low amount of compound available (see response to reviewers).
This makes me think that compost is very expensive and/or difficult to obtain - this is a serious problem;
Furthermore, the initial screening was performed with only two concentrations 10 and 50. This is insufficient.
In the literature there are thousands of molecules - natural, derived or synthetic - with more expressive activity and with much more data about their activities.
I reiterate that just one cytotoxicity study (MTT) is absurdly preliminary and different from what the authors defend, it is not enough for anything.
For this study to be considered in a journal of high impact and recognition as a molecule, the minimum is that the activity of the compounds associated with some chemotherapeutic already used in the clinic is also presented to us, in addition to other mechanism data or even in vivo studies (ie Erlich's Tumor).